# Prevalence, Diagnosis and Improving the Effectiveness of Therapy of Mastitis in Cows of Dairy Farms in East Kazakhstan

**DOI:** 10.3390/vetsci9080398

**Published:** 2022-07-30

**Authors:** Nurzhamal Mukhamadieva, Mardan Julanov, Dinara Zainettinova, Vasyl Stefanik, Zhanat Nurzhumanova, Aitbek Mukataev, Anuarbek Suychinov

**Affiliations:** 1Veterinary Department, Shakarim University, Semey 071412, Kazakhstan; nur71157@mail.ru (N.M.); tnt_rani@mail.ru (D.Z.); zhanat1970s@mail.ru (Z.N.); aitbek_mukataev@mail.ru (A.M.); 2Faculty of Veterinary, Kazakh National Agrarian Research University, Almaty 050000, Kazakhstan; mardan_58@mail.ru; 3Department of Obstetrics, Gynecology and Biotechnology of Animal Reproduction, Lviv National University of Veterinary Medicine and Biotechnology Named after S.Z. Gzhitsky, 79000 Lviv, Ukraine; stefanyk@bigmir.net; 4Kazakh Research Institute of Processing and Food Industry (Semey Branch), Semey 071410, Kazakhstan

**Keywords:** udder, mastitis, etiology, diagnosis, treatment

## Abstract

**Simple Summary:**

Mastitis is an inflammation of the mammary gland that occurs in response to adverse mechanical, physical, chemical and biological factors. The use of antibiotics as a treatment has a negative impact on milk quality. This paper proposes the treatment of mastitis with a drug that contains natural, environmentally friendly and pharmacologically active preparations. The inclusion of the preparation into the scheme of treatment allows to treat effectively inflammatory processes in the udder of cows and restore their productivity with minimal time and expenses. The drug has anti-inflammatory, antimicrobial and stimulating healing effects.

**Abstract:**

In the present work, the prevalence, etiological factors and effective treatment scheme of mastitis in cows of dairy farms “Balke” and “Madi-R” in Eastern Kazakhstan were investigated. In total, 210 heads were investigated on two farms. The incidence of mastitis in cows on dairy farms is not the same in different years. Average clinical mastitis was detected in 35.4% of cows in 2016, 19.6% in 2017, 28.5% in 2018, and in 2019 in 16.4% of cows. The prevalence rates of subclinical mastitis by year had some differences. So, in 2016—36.5% of cows, then in 2017—21.5%, 2018—19.3% and in 2019—22.6%. In cows with udder inflammation, serum calcium 9.37 ± 0.15 mg/% with a range of 8.0 to 10.8 mg/%, phosphorus 3.58 ± 0.07 mg/% (3.0 to 4.3 mg/%), reserve alkalinity 363.46 ± 6.69 mg/% (320 to 440), carotene 0.49 ± 0.03 mg/% (0.220 to 0.988 mg/%), which are in the lower limit of physiological parameters. The drug “Dorob” was tested during the study of comparative effectiveness of treatment methods. The results of the study showed that this drug has anti-inflammatory, antimicrobial and stimulating healing actions. The treatment of the sick cows with catarrhal mastitis has shown that a total of 8 cows have recovered in the control group and 10 cows in the experimental group with the preparation “Dorob”. The period of recovery in the control group was 8.8 ± 0.39, and in the experimental group—6.2 ± 0.28 (*p* < 0.05). The drug does not contain antibiotics and hormonal preparations. The inclusion of the drug in the scheme of treatment allows for effectively treating inflammatory processes in the udder of cows and restoring their productivity with minimal cost of time and money.

## 1. Introduction

The main sector of agriculture in Kazakhstan, including the East Kazakhstan region, is cattle breeding, and its development is conditioned by environmental, climatic and geographical conditions. The high number of livestock on large farms and complexes, and mechanization of basic production processes, including milking, revealed a number of critical problems in the prevention of serious diseases. A specific place among diseases is occupied by mammary gland diseases, which significantly affect animal productivity and reduce the quality of milk [1,2].

Mastitis in cows is an inflammation of the udder that causes pus and results in swelling and redness of the cow’s mammary gland. This disease can cause significant damage to large-scale milk production facilities [3,4]. For example, mastitis is estimated to cost the U.S. dairy industry approximately $2 billion in economic harm per year [5].

Udder inflammation occurs with violations of the rules of machine milking and poor conditions of housing and feeding of cows, which can facilitate the penetration and development of microbes in the mammary gland [6,7]. Mastitis can be caused by various reasons, primarily by inadequate or insufficient feeding, poor husbandry, improper housing and handling of animals, careless organization and performance of artificial insemination, due to various diseases of the reproductive organs, which appear most often during calving and in the postpartum period [8,9]. When veterinary and sanitary rules for maintenance of milking facilities and milking hygiene are violated, as well as the general resistance of the animal and mammary gland being reduced, microorganisms colonize the teat head and the outer opening of the teat canal. Then, microorganisms penetrate into the teat and suprapapillary cisterns (sinuses) and cause an infection process of varying severity from subclinical to clinically expressed inflammation of various forms [10].

Mastitis is a widespread disease in dairy complexes and farms. The incidence in some herds can reach 35–60% and up to 60–75% of cows in a herd can be affected within a year. Mortality rates from mastitis can in some cases exceed those from other diseases. During the disease period and after clinical recovery, the natural loss of milk per cow averages from 10 to 15% of the annual milk yield [11,12].

Mastitis can be caused by more than 140 different species of microorganisms. The most common of them can be grouped into the following groups: contagious (infectious), opportunistic and pathogenic microbes of the animal organism and environmental microorganisms [13].

Mastitis is categorized as clinical and subclinical mastitis. During the milking of cows with clinical mastitis, the properties of milk deteriorate (flakes, clots, watery or discolored milk) or the presence of swollen or indurated mammary quarters before milking [14]. During subclinical mastitis, an udder infection shows no external changes. Subclinical mastitis can result in an elevated milk somatic cell count and decrease milk yield, changing the composition of milk to influence milk quality. The somatic cell count (SCC) increases from about 200,000 SC/mL of milk (uninfected) to over 300,000 SC/mL of milk from mastitis cows [15,16].

In addition, data from many scientists show that not only staphylococci, streptococci and Escherichia coli cause mastitis, but also coronobacteria, salmonellae, mycobacteria, mycoplasmas, viruses, and fungi [17,18,19].

Getting quality and safe cow milk is an important task for dairy farms. Most scientists claim that mastitis in cattle is mainly caused by biological factors and therefore consider it an infectious disease of the mammary gland [20,21,22].

For treatment of mastitis, preparations containing (1) antiseptics (iodine, revanol, etc.), which are substances of non-selective action on pathogens, and (2) chemotherapeutic agents, which are substances of selective action on certain types of microorganisms are used. This group includes: sulfonamides, antibiotics, nitrofurans and new drugs of different groups (quinolones, quinoxaline derivatives, etc.) [23].

The main treatment for mastitis is usually administered by infusion of intramammary ointment or intramuscular or intravenous injections of antibiotics such as streptomycin, ampicillin, cloxacillin, penicillin and tetracycline [24,25]. However, treatment is expected to become problematic in the near future because of the rapid growth of antibiotic-resistant pathogens [26,27,28]. Therefore, it is necessary to introduce a set of effective preventive steps in the technology of milk production to control mammary gland diseases in cows, in particular, mastitis.

Drugs for intracisternal application in mastitis of cows: mastisan A, mastisan B, mastisan E, masticide, etc. [29,30]. They contain various combinations of antibiotics or antibiotics in combination with sulfonamides and oil bases. Usually, the oil base of the drugs prevents the contact of antibiotics with mastitis pathogens and their penetration into udder tissues. At the same time, about 50% of the antibiotics in the drug are excreted with milk, which creates environmental problems and limits the use of milk obtained from cows treated with such drugs [31,32].

Antimastitis drugs based on 3rd and 4th generations of cephalosporins have the shortest excretion period—no more than 60 h (for comparison, drugs based on lincomycin or gentamicin are excreted in five to seven days), and injectable drugs of the 3rd generation of cephalosporins (preparations based on ceftiofur) have no milk waiting period [33].

“Dioxilin gel” contains a complex that includes antibiotics and antiseptics in low concentrations. This makes it possible to fight effectively against the inflammatory process in the mammary glands of cows. The drug is a polymeric hydrogel, when applied to the udder of the animal, it dries quickly. The film that forms after drying ensures the long-lasting effect of active ingredients on inflammation. In addition, it protects against harmful external influences. “Dioxilin gel” is not toxic and has no effect on the quality of milk [34].

It is known in medicine and veterinary medicine sea that buckthorn oil contains vitamins A provitamin (carotene) 0.9–10.9 mg%, B1 (thiamine) 0.016–0.085 mg%, B2 (riboflavin) 0,030–0, 056 mg%, B3, B6, B9, K, macro-and micronutrients, folic acid 0.79 mg%, vitamin C (ascorbic acid) 54–316 mg%, P (bioflavonoids) 75–100 mg%, vitamin K (phylloquinones) 0.9–1.5 mg%, E (tocopherol) 8–18 mg. Sea buckthorn oil is recommended for preventing the development of inflammatory processes and has a protective effect on mucous membranes [35].

It is known that the method for the treatment of mastitis in cows, including biologically active components and diluting medium, in which, according to the invention, Vivaton—veterinary 10%; ASD-2f—4%; “Bursanatal”—liquid immunomodulator made from broiler Fabrica bursa 6%, and 0.9% isotonic solution is used as diluting medium [36]. The disadvantage of the prototype is its multicomponent composition, as well as the complexity of implementation of the invention. Virbac New Zealand Limited produces mastitis treatment preparation called Masticillin, which contains procaine penicillin. It is used to treat early season, multi-quarter mastitis, particularly Strep. uberis [37].

The main method of udder disease prevention is a science-based, cost-effective livestock breeding while complying with zoohygienic norms of maintenance, feeding and milking, providing a high level of natural resistance of the organism and animal productivity.

The purpose of the work is to determine the incidence, etiological factors and development of effective treatment of mastitis in cows on dairy farms in Eastern Kazakhstan. The practical significance of the work is the development of a drug for the treatment of mastitis in cows, which is easy to prepare and use. The drug does not contain antibiotics and hormonal drugs, allowing the effective treatment of mastitis in cows and restoring their productivity with minimal time and money. The drug does not require special expensive imported materials and expands the range of domestic mastitis drugs.

## 2. Materials and Methods

### 2.1. Experimental Base of the Study

The work was carried out as part of the research work of applied scientific research in the field of agriculture for 2018–2020 (0.0879) on the scientific and technical program: “Increasing the effectiveness of breeding methods in cattle breeding “on the project: “Development of effective breeding methods in dairy cattle breeding industry “on the measure: “Improvement of reproductive ability of dairy cows in the southern region”.

The experimental base of the study was the cattle farm “Balke” of the Beskaragai district and the cattle farm “Madi-R”, of the Znamenski rural district of Semey in East Kazakhstan region of the Republic of Kazakhstan. Cows in these farms were tested for subclinical and clinical forms of mastitis in all seasons of the year in the period from December 2016 to November 2019.

Peasant farm “Balke” of vegetable and dairy production is actively developing in the field of agriculture. Today the farm has more than 500 heads of black-motley breeding cows of the Kostroma breed (Figure 1). The total land area of the farm “Balke” is 2200 hectares (Table 1). In the farm “Balke” breeding stock of cattle: dairy, black-motley, Holstein–Friesian breeds.

For the studies, 210 milking cows of the black-motley breed at the age of 5–6 years with a live weight of 450–500 kg from the farm “Balke” and milking cows of the local Kazakh white-headed breed at the age of 5–6 years with a live weight of 350–470 kg from the farm “Madi-R” were selected.

### 2.2. Stages of Research and Treatment of Cows

The research was conducted from December 2016 to November 2019. The work was conducted in two stages. In the first stage, diagnostic research was carried out. In the second, the scheme of treatment was developed, and the results obtained during the experimental work were analyzed, checked and clarified.

During the study, the following methods were used: anamnesis collection, and clinical and laboratory methods of investigation. A medical history was created for each animal.

To study the effectiveness of the treatment protocols and testing the drug “Dorob” sick cows (20 animals) diagnosed with catarrhal mastitis were divided into two groups.

The first group consisted of cows treated with traditional methods of treatment (control group of animals, 10 cows). The treatment protocol of the control group of cows: the drug Mastiet forte, also cows of this group used the drug Ketoprof as an anti-inflammatory agent intramuscularly. The application of the above remedies was combined with biologically active supplements—Healthivit, which was injected subcutaneously in a dose of 6 mL once (Table 2).

The second group consisted of cows treated with the drug “Dorob” (10 cows). Cows in the second group were treated with the drug “Dorob” (Table 3). The drug “Dorob” was used intracisternally in a dose of 5 mL, twice a day with an interval of 10–12 h after completion of milking. The composition of this preparation: sea buckthorn oil in a ratio of 10:1 with ASD fraction 2. This preparation is an orange transparent oily liquid. Sea buckthorn oil, as well as ASD fraction 2, have vasoconstrictor, antiseptic, anti-inflammatory and analgesic effects and promote tissue regeneration. In addition, ASD fraction 2 as a biogenic stimulant increases the protective function and increases the body’s resistance to pathogenic and toxic influences. ASD fraction 2 contains low molecular weight organic compounds, including lower carboxylic acids, their amides and ammonium salts, choline esters of carboxylic acids, choline, primary and secondary amines, peptides, as well as inorganic nitrogenous compounds (salts of ammonium carbonate, ammonium acetic acid) and water [38].

As stated above, cows with catarrhal mastitis were divided into two groups (Figure 2). Animals of the first group were treated according to the scheme presented in Table 1. That is Mastiet forte—intracisternally for five days at a dose of 10 mL 2 times a day for five days. After injecting the drug, a slight massage of the affected quarter of the udder was carried out to distribute the drug evenly.

Moreover, cows of this group were treated with the drug Ketoprof as an anti-inflammatory agent intramuscularly. The dose of the drug Ketoprof—3 mL per 100 kg of animal weight once a day for three days. Application of the above drugs was combined with a complex vitamin-containing preparation—Healthivit subcutaneously in a dose of 6 mL once a day.

The therapeutic efficacy of the applied treatment protocols was determined by the number of cows recovered, the number of days of recovery, and the manifestation of relapses. Economic efficiency was determined by taking into account the cost of drugs, labor costs of veterinarians and service personnel, and the prevented damage from the loss of dairy products.

### 2.3. Testing for Mastitis

Diagnosis was based on the classification of mastitis according to A.P. Studentsov [36]. To detect subclinical mastitis the following tests were used: bromothymol, mastidine, dimestine, Whiteside, sedimentation of milk, California mastitis test. Furthermore, in work the device of express mastitis diagnosis (PEDM), Lactan was used. Lactan 4,1-mini device was used to check the amount of dry skimmed milk residue, density, the water–fat ratio in milk, and the number of somatic cells, and the Miltek device was used to determine mastitis milk.

### 2.4. Biochemical Blood Tests

To determine the number of hemoglobin, erythrocytes and leukocytes in the blood we conducted research using a PCE 90 Vet hematology analyzer. A hematological analyzer is a device for determining the number of blood cells in animals and the amount of hemoglobin for diagnosis in veterinary laboratories. The modern apparatus is very convenient and fast, it takes a few minutes to carry out the analysis, and the response time of the general analyzers is 7–10 min.

Hemanalyzer standard reagents are intended for the analysis of 170–200. Defined parameters: erythrocytes (RBC), hemaglobin (HGB), hematocrit (HCT), white blood cells (WBC), mean erythrocyte volume (MCV), mean hemaglobin volume per erythrocyte (MCH), mean hemaglobin concentration per erythrocyte (MSHC), erythrocyte volume ratio change (RDW %), platelets (PLT), mean platelet volume (MPV), thrombocrit (PCT), platelet length particles (PDW). The histogram displays the volume of erythrocytes, white blood cells, and platelets.

### 2.5. Determination of the Chemical Composition of the Feed

Moisture content was determined according to GOST 13496.3-92 [39]. Opened beakers with feed are placed in a drying cabinet preheated to a temperature of (130 ± 2) °C. Drying is carried out for 40 min, counting from the moment when the temperature reaches 130 °C.

After drying and cooling the beakers with feed are weighed to the second decimal place.

The mass fraction of moisture (W) in percent is calculated according to the formula
$$W=\frac{m_{1}-m_{2}}{m_{1}-m}\times 100$$
where
*m*_1_ is the mass of the beaker with the sample before drying, g;*m*_2_—mass of the beaker with the sample after drying and cooling, g;*m*—mass of empty beaker, g.


### 2.6. Determination of Carotene in Feeds

The essence of the method consists of the extraction of carotene with petroleum ether or gasoline and photometric measurement of the color intensity of the extract, which depends on the carotene content [40].

### 2.7. Determination of the Reserve Alkalinity of Blood by Kondrakhin’s Method

Principle of the method. In one half of the flask, blood plasma is treated with sulfuric acid, which results in the release of carbon dioxide, which is part of the bicarbonates. The released carbon dioxide is absorbed by a sodium hydroxide solution, which is in the other half of the flask. The excess sodium hydroxide that was not reacted with carbon dioxide and half of the sodium hydrocarbonate (NaHCO_3_) formed during the absorption of CO_2_ is titrated with a sulfuric acid solution. The amount of initially bound sodium hydroxide is used to determine the amount of carbon dioxide released from the plasma, which is equivalent to the bicarbonate content [41].

### 2.8. Statistics

Each experiment was performed in triplicate and values were expressed as mean ± SEM of three independent observations. Statistical analysis was performed using Statistica 12.0 (STATISTICA, 2014; StatSoft Inc., Tulsa, OK, USA). The differences between samples were evaluated using one-way ANOVA. The Tukey HSD test was used for means comparisons. A *p*-value < 0.05 was considered statistically significant.

## 3. Results

### 3.1. Incidence of Mastitis in Cows on Dairy Farms of Balke and Madi-R

The incidence of mastitis in cows on dairy farms of Balke and Madi-R farms showed different incidences in different years (Table 4). A total of 210 cows were examined on two farms. On average, clinical mastitis was registered in 35.4% of cows in 2016, 19.6% in 2017, 28.5% in 2018 and 16.4% in 2019. Furthermore, the prevalence rates of subclinical mastitis had some differences by year. For example, in 2016 it was 36.5%, then in 2017—21.5%, in 2018—19.3 and in 2019—22.6%.

Out of 90 cows, 56 cows had clinical mastitis, which amounted to 35.4%. Subclinical mastitis was present in 34 cows, which amounted to 36.5%. Catarrhal mastitis occurred in 24 cows or 32%.

Analyzing the data in Table 5, we revealed the prevalence of subclinical forms of mastitis. In winter, the rate of subclinical mastitis gradually increased. In spring, there was a dramatic rise. A decline was observed in summer, and a high level of subclinical mastitis occurred in the autumn season. The incidence of subclinical mastitis in cows was 15 animals in winter, 29 animals in spring, 19 animals in summer, and 30 animals in fall.

According to the data (Table 6), the highest number of clinical mastitis occurs in the spring and autumn periods. Clinical mastitis in winter was detected in 38 animals, in spring—48 animals, in summer—27 animals, in autumn—45 animals.

Our research showed the occurrence by season of catarrhal mastitis in 2016 was—32%, in 2017—21.3%, in 2018—25.3%, and in 2019—21.3% (Table 7). The rate of catarrhal mastitis gradually increased during the winter and spring seasons. There was a gradual decline in the summer. The increase in clinical mastitis was observed in autumn.

The incidence of mastitis in cows was 15 head in winter, 19 head in spring, 10 head in summer, and 31 head in fall.

### 3.2. Feeding Diets of Cows

Moreover, when identifying predisposing factors of mastitis, we studied the feeding diets of cows in the summer-autumn and winter-spring periods of the year. So, in the summer-autumn period, animals were on pasture—grass pasture hay (steppe, meadow), additional feeding, salt, chalk, water at will, mixed fodder 3.5 kg per head during milking.

During the winter period, the cows’ ration consisted of 25 kg of silage, 10 kg of mono-fodder, 5 kg of hay, 3.5 mixed fodder, 1.5 kg of sunflower meal, 4 kg of vitamin-mineral additive, 150 g of molasses, 25 g of chalk, 60 g of salt. The daily ration of the first heifers was 14.6 fodder units (Table 8).

The diet of the first heifers in the summer-autumn periods consisted of 20 kg of silage, 10 kg of mono-fodder, 8 kg of hay, and 350 g of mixed fodder. Per liter of milk, 1.5 kg of sunflower meal, 3 kg of vitamin-mineral additive, 100 g of molasses, 25 g of chalk, and 50 g of salt. The daily ration was 8.8 fodder units.

The analysis of our studies showed that in the summertime, with sufficient proteins, carbohydrates, vitamins and minerals in green feed, mastitis diseases in cows are less common.

Chemical analysis of forages was characterized by the following indicators (Table 9). Carotene content in hay was 8.67 ± 1.60 mg/kg (with variation from 4.2 mg/kg to 10.3 mg/kg). The carotene content of the silage was 5.30 ± 2.15 mg/kg (ranging from 2 mg/kg to 8 mg/kg). The crude protein level in the silage was 20.60 ± 3.47 g/kg (15.6 to 25.4 g/kg). Digestible protein content in hay was 50.50 ± 4.08 g/kg (39.1 to 64.2 g/kg) and in silage was 11.00 ± 2.50 g/kg (7.2 to 14.2 g/kg). Calcium and phosphorus in hay contained between 4.48 ± 0.73 g/kg (1.7–6.2 g/kg and 0.92–2.00 g/kg), and in silage their content was 0.56 ± 0.15 g/kg (0.326 to 0.757 g/kg and 0.38 ± 0.12 g/kg (0.236 to 0.565 g/kg), respectively.

The results of the analysis of fodder of the farm “Balke” of the Beskaragai district in the East Kazakhstan region showed that the nutrient value of hay was on average 0.35 ± 0.04 f.u./kg (from 0.35 to 0.58 f.u./kg). The results of the study of hay and silage are presented in Table 8.

### 3.3. Analysis of Cow’s Blood

As it is known, in mastitis, the pathological process is not limited to the tissues of the affected quarter of the mammary gland, it covers the whole organism, which as a whole participates in the elimination of the disease [42,43]. Accordingly, during inflammation of the mammary gland, there are biochemical changes in the secretion from the affected udder, as well as in blood. These changes may occur even before the clinical manifestation of mastitis, therefore, having great diagnostic value in practical veterinary medicine [44,45].

We also examined blood serum from 26 cows with mastitis for some biochemical parameters, specifically for calcium, phosphorus, reserve alkalinity and carotene content (Table 10). If we take into account the fact that normally the blood of cows should contain in mg/% of calcium—11.3–13.1, phosphorus—4.0–4.6, reserve alkalinity—460–540, carotin—0.5–2.28, then according to the research these indices were reduced or were at the lower limit of the physiological characteristics in the blood of the cows with mastitis.

In cows with udder inflammation, serum calcium was found to be 9.37 ± 0.15 mg/% with a range of 8.0 to 10.8 mg/%, phosphorus 3.58 ± 0.07 mg/% (3.0 to 4.3 mg/%), reserve alkalinity 363.46 ± 6.69 mg/% (320 to 440), carotene 0.49 ± 0.03 mg/% (0.220 to 0.988 mg/%).

Cows with mastitis were divided into two groups for treatment. Before treatment, we determined the level of erythrocytes (RBC), hemoglobin (HGB), hematocrit (HCT), leukocytes (WBC), mean erythrocyte volume (MCV), mean hemoglobin volume in erythrocyte (MCH), mean erythrocyte hemoglobin concentration (MCHC), change in erythrocyte volume ratio (RDW %), platelets (PLT), mean platelet volume (MPV), thrombocrit (PCT), platelet distribution width (PDW).

The results of blood tests of cows with mastitis before treatment were (Table 11): White blood cell (WBC)—9.86 ± 1.59 × 10^9^/L, red blood cell (RBC)—6.50 ± 0.39 × 10^12^/L, hemoglobin (HGB)—79.50 ± 1.83 g/L, hematocrit (HCT)—17.79 ± 0.49%, mean red blood cell volume (MCV)—29.11 ± 0.64 fL, mean hemoglobin per red blood cell (MCH)—12, 27 ± 0.55 pg, mean erythrocyte hemoglobin concentration (MCNS)—417.60 ± 5.28 g/L, change in erythrocyte volume ratio (RDW %)—15.67 ± 0.34%, platelet (PLT)—240.10 ± 24.02 × 10^9^/L.

According to the results of the study (Table 12) the blood of cows with catarrhal mastitis was as follows: leukocyte (WBC) in the experimental group was: −13.88 ± 1.80 × 10^9^/L, in the control group—8.99 ± 0.47, erythrocyte (RBC)—8.10 ± 0.73 × 10^12^/L, in the control group—7.89 ± 0.46. Hemoglobin (HGB)—98.40 ± 4.58 g/L, in the control group 133.20 ± 31.94. Hematocrit (HCT)—24.44 ± 1.10%, in controls—35.06 ± 1.60. Mean erythrocyte volume (MCV)—31.66 ± 1.04 fL, in controls—46.88 ± 1.36. Mean erythrocyte hemoglobin volume (MCH)—11.06 ± 1.09, in controls 15.45 ± 0.43 pg. Mean erythrocyte hemoglobin concentration (MSHC)—413.80 ± 5.28 g/L, change in erythrocyte volume ratio (RDW %)—15.53 ± 0.29%, platelet (PLT)—243.50 ± 27.20 × 10^9^/L.

As the results of the treatment of sick cows with catarrhal mastitis (Table 13) showed, 8 cows in the control group and 10 cows in the experimental group recovered.

There were two cows that did not recover. Terms of recovery in the control group were 8.8 ± 0.39, in the experimental group 6.2 ± 0.28 (*p* < 0.05). Two cases of relapses in the control group were detected.

Thus, during the comparative effectiveness of the treatment methods, the drug “Dorob” proved to be effective. The drug had anti-inflammatory, antimicrobial and stimulating healing effects. It should be noted that the drug did not contain antibiotics and hormones. Therefore, the inclusion of the drug in the scheme of treatment allows, with minimal time and money, effectively treating inflammatory processes in the udder of cows, restoring their function and productivity.

## 4. Discussion

Despite the constant improvement of methods to prevent mastitis, mammary gland inflammation remains the most common disease in cows on dairy farms and cattle complexes to this day [46,47,48,49]. In our research, we found out that mastitis in conditions of the East Kazakhstan region is widespread. The main causes of mastitis are inadequate feeding, violation of keeping conditions, milking improprieties, udder trauma and a variety of stresses [50].

We also studied the occurrence of subclinical and clinical mastitis by seasons of the year for the period 2016–2019. Clinical mastitis was recorded in 2016 in 35.4% of cows, in 2017 in 19.6%, 2018 in 28.5% and in 2019 in 16.4%. Furthermore, the prevalence rates of subclinical mastitis had some variation by year. Thus, in 2016—36.5%, in 2017—21.5%, 2018—19.3 and in 2019—22.6%.

Analyzing the data of the conducted study, it can be concluded that high levels of subclinical and clinical mastitis are revealed in the autumn time of the year.

The prevalence of mastitis among cows in different years is not the same. The results of the data show that with low air temperature and high precipitation the incidence of mastitis among cows increases. At the beginning of spring (draughts, coldness, dampness), seasonal mastitis outbreaks are observed. Thus, the deterioration of weather conditions negatively affects milking animals, which leads to an increase in the number of all forms of mastitis in cows [51,52,53]. (Cervinkova et al., 2013) studied that half of the collected milk from 16 dairy farms in the Czech Republic was contaminated by microorganisms with the potential to cause mastitis [54]. In the United Kingdom, the annual prevalence of clinical mastitis has been estimated at 35% of the total dairy cow population [55].

The most common forms of mastitis were subclinical and catarrhal mastitis [56,57]. Therefore, in our work, we selected animals with the catarrhal form of mastitis to determine treatment efficacy. The feeding diets of cows in the summer-autumn and winter-spring periods of the year were studied for the manifestation of mastitis. Analysis of our studies showed that in summer, with sufficient proteins, carbohydrates, vitamins and minerals in green fodder, mastitis diseases in cows occur less frequently.

The results of the analysis of forages of the farm “Balke” showed that the nutritive value of hay on average was 0.35 ± 0.04 f.u./kg.

We also examined blood serum from 26 cows with mastitis for some biochemical parameters, namely calcium, phosphorus, reserve alkalinity and carotene.

Such a violation of mineral metabolism, especially in combination with protein and vitamin deficiencies in the diet, in our opinion, leads to a decrease in the resistance of the body of cows and subsequently predisposes the body to the manifestation of pathological processes, including mastitis.

To treat cows with catarrhal mastitis they were divided into two groups.

According to the results of blood tests of cows with catarrhal mastitis before treatment, it was found: White blood cell (WBC)—9.86 ± 1.59 × 10^9^/L, red blood cell (RBC)—6.50 ± 0.39 × 10^12^/L, hemoglobin (HGB)—79.50 ± 1.83 g/L, hematocrit (HCT)—17.79 ± 0.49%, mean red blood cell volume (MCV)—29.11 ± 0.64 fL, mean hemoglobin per red blood cell (MCH)—12.27 ± 0.55 pg, mean erythrocyte hemoglobin concentration (MCNS)—417.60 ± 5.28 g/L, erythrocyte volume ratio change (RDW %)-15.67 ± 0.34%, platelet count (PLT)—240.10 ± 24.02 × 10^9^/L. Blood values were in the lower limit of physiological indices.

The results of blood analysis of cows with catarrhal mastitis revealed: that white blood cell (WBC) in the experimental group was: −13.88 ± 1.80 × 10^9^/L, in the control group—8.99 ± 0.47, erythrocyte (RBC)—8.10 ± 0.73 × 10^12^/L, in the control group—7.89 ± 0.46. Hemoglobin (HGB)—98.40 ± 4.58 g/L, in the control group 133.20 ± 31.94. Hematocrit (HCT)—24.44 ± 1.10%, in controls—35.06 ± 1.60. Mean erythrocyte volume (MCV)—31.66 ± 1.04 fL, in controls—46.88 ± 1.36. Mean erythrocyte hemaglobin volume (MCH)—11.06 ± 1.09, in controls 15.45 ± 0.43 pg, Mean erythrocyte hemaglobin concentration (MSHC)—417.60 ± 5.28 g/L, change in erythrocyte volume ratio (RDW %)—15.67 ± 0.34%, platelet (PLT)—240.10 ± 24.02 × 10^9^/L.

Alhussien et al. (2015) observed that the blood and milk lymphocytes and monocytes of cows with mastitis were significantly decreased [58].

One of the tasks of this work is to develop treatment protocols for cows with catarrhal mastitis. For this purpose, we selected such a scheme, which, firstly, would be simple, easily performed in production conditions, secondly, would not contain antibiotics and hormonal agents, and, thirdly, would be effective both therapeutically and economically.

Such a remedy was the drug “Dorob”. This remedy for the treatment of cows with mastitis included ASD fraction 2 and sea buckthorn oil. The ratio of ASD fraction 2 to sea buckthorn oil was 1:10.

Studies of compatibility of the drug components on the animal body, as well as the pharmacotherapeutic effect in the treatment of mammary gland (mucous membranes) diseases, were carried out. Materials used in the experiments were: egg white, fresh milk, sea buckthorn oil, and ASD fraction 2. We studied the interaction of milk and sea buckthorn oil, then egg albumen with sea buckthorn oil, then egg albumen with ASD fraction 2, and a water-milk solution in the ratio 1:10 with ASD fraction 2. There were no flakes or clots formed in the udder secretion. The solution turned out homogeneous, dispersedly thick.

Abboud M. et al. (2015) reported that essential oils from *Thymus Vulgaris* and *Lavandula Angustifolia* showed strong antibacterial activity against the main strains of mastitis Staphylococcus and Streptococcus [59]. Dilshad et al. (2010) analyzed the practices for mastitis treatment and reported that 25 different plant species were being used for the treatment and control of mastitis in cows and buffaloes in Pakistan [60]. Pașca et al. (2020) proposed the treatment of mastitis caused by Gram-positive bacteria with natural products, containing different plant extracts and propolis [61].

From the data, we can conclude that the fluids obtained in the interaction of the drug components with the animal’s body have a high viscosity. Chemical processes in their environment proceed slowly, diffusion and resorption are delayed, the fluids have adsorptive properties, decrease the sensitivity of nerve endings and promote faster healing of inflammatory processes.

## 5. Conclusions

Analyzing the data of the study we can conclude that a high level of subclinical and clinical mastitis is revealed in autumn time of the year. The main reasons for mastitis in milking cows are poor quality of feeding affected by mold fungi and incomplete feeding, violation of keeping conditions and milking inaccuracies. The inclusion of the preparation “Dorob” into the scheme of treatment allows to treat effectively inflammatory processes in the udder of cows and restore their productivity with minimal time and expenses. The drug has anti-inflammatory, antimicrobial and stimulating healing effects. The product does not contain antibiotics and hormones.

## Figures and Tables

**Figure 1 vetsci-09-00398-f001:**
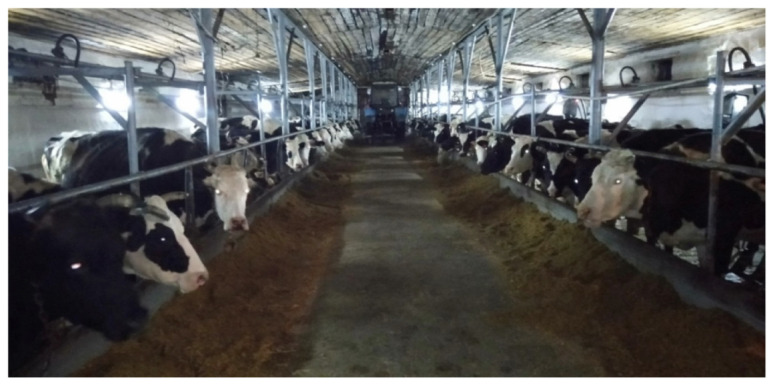
Cows of the “Balke” farm.

**Figure 2 vetsci-09-00398-f002:**
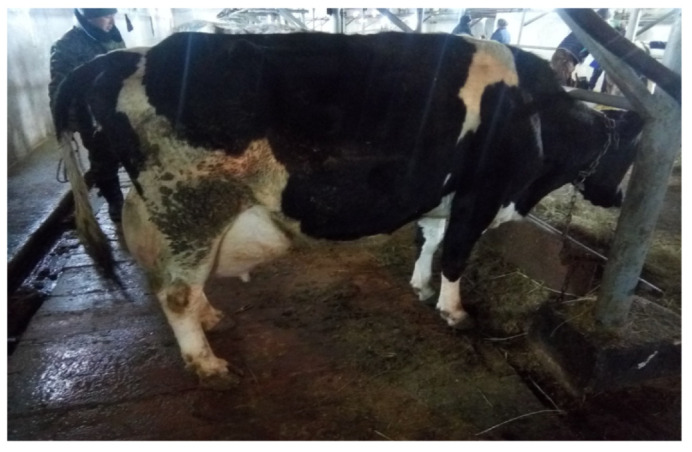
Cow with mastitis.

**Table 1 vetsci-09-00398-t001:** Average annual milk of the farm “Balke”.

Number of Milking Cows (Heads)	Time (days) Milked During the Year	Milk from One Cow per Day (Liters)	Annual Milking per Cow (Liters)	Annual Milking of Milk from All Cows (Tons)
200	305	20	6100	1220

**Table 2 vetsci-09-00398-t002:** Treatment protocol for control group cows with catarrhal mastitis.

Product Name	Application	Single Dose	Treatment Days				
			1	2	3	4	5
Mastiet Forte	Intracisternally, twice a day at 10–12 h intervals.	1 syringe dispenser	+	+	+	+	+
Ketoprof	Intramuscularly	3 mL per 100 kg	+	+	+		
Healthivit	intramuscular	6 mL	+				

**Table 3 vetsci-09-00398-t003:** Treatment protocol for cows of the experimental group with catarrhal form.

Name of the Drug	Application	Dose	Treatment Days				
			1	2	3	4	5
“Dorob”	Intracisternally twice a day at intervals of 10–12 h.	5 mL per quarter of the udder	+	+	+	+	+

**Table 4 vetsci-09-00398-t004:** Analysis of the incidence of mastitis among cows of Balke and Madi-R farms.

Diseases of Cows	2016		2017		2018		2019	
	Number of Cases	%	Number of Cases	%	Number of Cases	%	Number of Cases	%
Clinical mastitis	56	35.4	31	19.6	45	28.5	26	16.4
Including:								
Serous mastitis	13	14.4	8	15.6	13	20.6	5	10.6
Fibrinous acute mastitis	16	17.7	4	7.8	7	11.1	2	4.2
Catarrhal mastitis	24	32	16	21.3	19	25.3	16	21.3
Hemorrhagic mastitis	1	1.1	3	5.8	5	7.9	2	4.2
Purulent mastitis	2	2.2		-	1	1.5	1	2.1
Subclinical mastitis	34	36.5	20	21.5	18	19.3	21	22.6
Total	90	100	51	100	63	100	47	100

**Table 5 vetsci-09-00398-t005:** Occurrence of subclinical mastitis in cows by season of the year for the period 2016–2019.

Year	Season of the Year									
	Winter		Spring		Summer		Autumn		Total	
	Number of Cases	%	Number of Cases	%	Number of Cases	%	Number of Cases	%	Number of Cases	%
2016	4	26.6	12	41.4	8	42.1	10	33.3	34	36.5
2017	3	20	6	20.7	5	26.3	6	20	20	21.5
2018	3	20	4	13.7	4	21	7	23.3	18	19.3
2019	5	33.3	7	24.1	2	10.5	7	23.3	21	22.6
Total	15	100	29	100	19	100	30	100	93	100

**Table 6 vetsci-09-00398-t006:** Occurrence of clinical mastitis in cows by season of the year for the period 2016–2019.

Year	Season of the Year									
	Winter		Spring		Summer		Autumn		Total	
	Number of Cases	%	Number of Cases	%	Number of Cases	%	Number of Cases	%	Number of Cases	%
2016	14	36.9	12	25	10	37	20	44.4	56	35.4
2017	5	13.1	9	18.7	6	22.2	11	24.4	31	19.6
2018	16	42.1	18	37.5	5	18.5	6	13.3	45	28.5
2019	3	7.8	9	18.7	6	22.2	8	17.8	26	16.4
Total	38	100	48	100	27	100	45	100	158	100

**Table 7 vetsci-09-00398-t007:** Occurrence of catarrhal mastitis of cows by seasons of the year for the period 2016–2019.

Year	Season of the Year									
	Winter		Spring		Summer		Autumn		Total	
	Number of Cases	%	Number of Cases	%	Number of Cases	%	Number of Cases	%	Number of Cases	%
2016	5	33.3	7	36.8	4	40	8	25.8	24	32
2017	3	20	5	26.3	2	20	6	19.3	16	21.3
2018	3	20	5	26.3	3	30	8	25.8	19	25.3
2019	4	26.6	2	10.5	1	10	9	29	16	21.3
Total	15	100	19	100	10	100	31	100	75	100

**Table 8 vetsci-09-00398-t008:** Feeding ration of cattle for the winter—stall period on the farm “ Balke”.

Groups of Animals	Unit	Fodder	Silage	Monofeed	Hay	Sunflower Meal	Vitamin Supplement	Treacle, g	Chalk, g	Salt, g	Feed Units
Breeding cows	kg	3.5	25	10	5	1.5	4	1000	25	60	12.1
Heifers at calving	kg	350 g per 1 L	20	10	8	1.5	3	1000	25	50	14.6
Dry cowbane	kg	2.0	10	5	8	-	-	5000	20	50	8.8

**Table 9 vetsci-09-00398-t009:** Results of analysis of hay and silage.

Name	Carotene, mg/kg	Ca, g/kg	P, g/kg	Protein, g/kg	Crude Protein, g/kg	Feed Units, kg/kg	Moisture, %
Hay	8.67 ± 1.60	4.48 ± 0.73	2.72 ± 1.39	50.50 ± 4.08	-	0.49 ± 0.04	-
Silage	5.30 ± 2.15	0.56 ± 0.15	0.38 ± 0.12	11.00 ± 2.50	20.60 ± 3.47	-	78.77 ± 1.56

**Table 10 vetsci-09-00398-t010:** Results of biochemical blood serum tests of cows with mastitis, *n* = 26.

Indicator	Number of Calvings	Results, mg%			
		Ca	P	Alkaline Reserve	Carotene
M ± m	4.88 ± 0.36	9.37 ± 0.15	3.58 ± 0.07	363.46 ± 6.69	0.49 ± 0.03

**Table 11 vetsci-09-00398-t011:** Results of blood tests before treatment.

Blood Parameters								
WBC	RBC	HGB	HCT	MCV	MCH	MCHC	RDW	PLT
9.86 ± 1.59 × 10^9^/L	6.50 ± 0.39 × 10^12^/L	79.50 ± 1.83 g/L	17.79 ± 0.49%	29.11 ± 0.64 fL	12.27 ± 0.55 pg	417.60 ± 5.28 g/L	15.67 ± 0.34%	240.10 ± 24.02 × 10^9^/L

**Table 12 vetsci-09-00398-t012:** Results of blood analysis after the test.

Blood Parameters									
Groups	WBC × 10^9^/L	RBC × 10^12^/L	HGB g/L	HCT %	MCV fL	MCH pg	MCHC g/L	RDW %	PLT × 10^9^/L
experimental, *n* = 10	13.88 ± 1.80	8.10 ± 0.73	98.40 ± 4.58	24.44 ± 1.10	31.66 ± 1.04	11.06 ± 1.09	413.80 ± 7.39	16.53 ± 0.29	243.50 ± 27.20
control, *n* = 10	8.99 ± 0.47	7.89 ± 0.46	133.20 ± 31.94	35.06 ± 1.60	46.88 ± 1.36	15.45 ± 0.43	344.40 ± 8.71	16.41 ± 0.40	311.00 ± 20.46

**Table 13 vetsci-09-00398-t013:** Results of treatment of cows with catarrhal mastitis.

Indicators	Groups	
	Control, *n* = 10	Experimental, *n* = 10
Cows recovered	8	10
Cows not recovered	2	-
Terms of recovery, days	8.8 ± 0.39	6.2 ± 0.28 *p* < 0.05
Manifestations of relapses	2	-

## Data Availability

All data generated or analyzed during this study are included in this published article.

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
