# Peer review of "Prevalence, Diagnosis and Improving the Effectiveness of Therapy of Mastitis in Cows of Dairy Farms in East Kazakhstan"

_vetsci, 2022, doi:10.3390/vetsci9080398_

Round 1
Reviewer 1 Report
The article entitled “Prevalence, diagnosis and improving the effectiveness of therapy of mastitis in cows of dairy farms in East Kazakhstan” reports research on the prevalence, etiological factors and effective treatment scheme of mastitis in cows of dairy farms "Balke" and "Madi-R" in Eastern Kazakhstan in 2016-2019. The investigation is extensive and exhaustive on the occurrence of the disease in cows from prevalence to treatment which is a great advantage of this work. In my opinion the studies are well designed and the results are performed in the right way. However, I have one issue for Discussion, which in my opinion should be changed. Could the authors compare their results with other authors? I am sure that a similar study was conducted before e.g Cervinkova (Veterinarni Medicina, 58, 2013 (11): 567–575), V Krömker (doi.org/10.1111/rda.13032), Oliveira (https://doi.org/10.3168/jds.2013-7756). Apart from this one but significant remark, which the authors should consider/improved, I have nothing more to add.
Author Response
Dear Reviewer All corrections and additions are highlighted in blue. We improved Introduction (added more information on mastitis), Materials (added more information about the studied area and farm) and Discussion (added related research works) and Reference section.

Reviewer 2 Report
An interesting topic of dairy farming was investigated. Unfortunately, a lot of information is missing in the whole article to be able to evaluate the results.
In the first chapter I would like to see much more information on mastitis and the possible causes. As mastitis is a multifactorial disease, more information is needed.
In chapter 2.1 the region, the farms and the cows are not described enough. Data on milk production in the country is missing to be able to classify the farm. The farms lack a description of the husbandry, the feeding, the milking technique,... in order to be able to classify the causes of mastitis. In the case of the cows, there is a lack of information on average milk yield, absolute milk yield, milk composition, number of lactations of the cows,...
There is no comparative literature analysis on the subject of alternative mastitis drugs.
There is much that is not comprehensible in the selection of cows for the study. How advanced was the mastitis when the treatment was started, what was the cause of the mastitis, how would the two groups be formed, what were the performance values of the cows, what was their lactation status, what was the general condition of the cows,...?
The two herds and both breeds were mixed in the study. This is scientifically not entirely correct.
The action pattern of the drug "Dorob" is not explained scientifically.
Chapter 2.8 actually says nothing about the statistics. More information is needed here. The explanation on animal welfare belongs in the corresponding footnote of the article.
How has the blood testing of the 26 cows been divided between the two groups.
The research is not compared with the international literature on the subject.
The discussion and summary is not scientific in its current form. The comparison with other studies is missing here.
Author Response

(The authors gave the same response as above.)
